# Tetrazine-Induced Bioorthogonal Activation of Vitamin E-Modified siRNA for Gene Silencing

**DOI:** 10.3390/molecules27144377

**Published:** 2022-07-08

**Authors:** Xueli Zhang, Amu Gubu, Jianfei Xu, Ning Yan, Wenbo Su, Di Feng, Qian Wang, Xinjing Tang

**Affiliations:** 1State Key Laboratory of Natural and Biomimetic Drugs, School of Pharmaceutical Sciences and Chemical Biology Center, Peking University, No. 38 Xueyuan Rd., Beijing 100191, China; juhao@bjmu.edu.cn (X.Z.); amu@bjmu.edu.cn (A.G.); xjf@bjmu.edu.cn (J.X.); 1716386027@bjmu.edu.cn (N.Y.); 1711210115@pku.edu.cn (W.S.); fengdi@bjmu.edu.cn (D.F.); qian.wang@bjmu.edu.cn (Q.W.); 2State Key Laboratory of Pharmaceutical Biotechnology, Nanjing University, Nanjing 210023, China

**Keywords:** caged siRNA, bioorthogonal activation, siRNA, gene silencing

## Abstract

The temporal activation of siRNA provides a valuable strategy for the regulation of siRNA activity and conditional gene silencing. The bioorthogonal bond-cleavage reaction of benzonorbonadiene and tetrazine is a promising trigger in siRNA temporal activation. Here, we developed a new method for the bio-orthogonal chemical activation of siRNA based on the tetrazine-induced bond-cleavage reaction. Small-molecule activatable caged siRNAs were developed with the 5′-vitamin E-benzonobonadiene-modified antisense strand targeting the green fluorescent protein (GFP) gene and the mitotic kinesin-5 (Eg5) gene. The addition of tetrazine triggered the reaction with benzonobonadiene linker and induced the linker cleavage to release the active siRNA. Additionally, the conditional gene silencing of both exogenous GFP and endogenous Eg5 genes was successfully achieved with 5′-vitamin E-benzonobonadiene-caged siRNAs, which provides a new uncaging strategy with small molecules.

## 1. Introduction

Small interfering RNA (siRNA) has been considered a powerful gene silencing tool for its sequence-specific degradation of mRNA [1]. For the spatial or temporal functional evaluation of target gene silencing, conditional activation of siRNAs is sometimes required. Currently, light is the most widely used exogenous stimuli for the activation of photocaged siRNAs for gene silencing [2,3,4]. In addition to light, some endogenous stimuli, including pH [5], enzyme [6], and glutathione [7,8,9,10], were reported to activate siRNA. However, caged siRNAs that were sensitive to these endogenous stimuli were usually heterogeneous, unstable and uncontrollable in cellular systems [11]. Additionally, these caged siRNA might be activated at an undesired time and positions or not efficiently activated. Thus, it is noteworthy to develop new methods of caging siRNAs that are biorthogonal, biocompatible and readily synthesized.

Bioorthogonal reactions are orthogonal to and compatible with biological systems, which has been emerging as a new research area in recent years [12,13]. To date, bioorthogonal bond-cleavage reactions have found many applications in various areas, including controlling the activity of proteins [14], drug delivery [15], nucleic acid-templated reactions [16], and others. These bond-cleavage reactions might also provide a unique way to activate siRNA with exogenous small molecules, which might overcome the limitations of other caged siRNAs [11]. Among various kinds of bioorthogonal bond-cleavage reactions, inverse electron demand Diels–Alder (IEDDA) reactions between tetrazine and dienophile are considered the most promising bioorthogonal reactions for in vivo application due to their fast reaction rate and the low toxicity of tetrazine [17,18,19]. Previously, the bioorthogonal reaction between benzonorbonadiene and tetrazine also provided the similar bond cleavage [20]. Considering the feasibility of synthesis and acid-stability of reactive moieties, the benzonorbonadiene moiety could be a more favorable group as the tetrazine sensitive linker based on the bioorthogonal bond-cleavage reaction between benzonorbonadiene and tetrazine.

Vitamin E was previously conjugated to oligonucleotides as a tissue-specific moiety (such as liver cells) for the delivery of oligonucleotide drugs [21,22,23]. However, vitamin E modification of the 5′ end of RNA oligonucleotides (esp. antisense RNA strand) impeded siRNA gene silencing activity for target mRNA [22,24,25]. According to our previous report on photolabile siRNAs, single vitamin E modification on the 5′ terminal of the siRNA antisense strand would fully block its gene silencing activity, which could be recovered upon light-induced photocleavage of the photolabile linker between vitamin E and RNA oligonucleotides [26]. Here, we chose vitamin E-benzonobonadiene-caged-siRNA for the proof-of-concept study to demonstrate the potential application of the biorthogonal uncaging strategy for masking and activation of siRNA gene silencing activity. These chemical stimuli-responsive vitamin E-caged siRNAs targeting both green fluorescent protein (GFP) gene and mitotic kinesin-5 (Eg5) gene were then designed and synthesized through a benzonorbonadiene linker (BL) between Vitamin E and the 5′ terminal of the siRNA antisense strand. Upon the addition of tetrazine, the bioorthogonal reaction of benzonobonadiene and tetrazine induced the breakage of benzonorbonadiene linker and released the active siRNAs for gene silencing of both endogenous and exogenous genes in cells.

## 2. Results and Discussion

### 2.1. Rational Design and Synthesis of Benzonorbonadiene Linker and Vitamin E-Benzonorbonadiene-Caged siRNA

To develop vitamin E-benzonorbonadiene caged siRNAs, we first synthesized benzonorbonadiene linker and vitamin E phosphoramidites. The benzonorbonadiene linker phosphoramidite was designed according to the literature report [20] with minor reversion, as shown in Figure 1. Intermediate compound **4** was synthesized from pyrrole-2-carboxaldehyde in three steps. After the monobromination of tyrosol and DMT protection of the hydroxyl group, the as-prepared derivative **7** was further silyted with 1,1,1,3,3,3-hexamethyldisilazane (HMDS), and lithiated with n-BuLi, followed by the reaction with sulfonyldiimidazole to afford o-(trimethylsilyl)aryl imidazolylsulfonated derivative **9**. Derivative **4** and derivative **9** underwent the cycloaddition reaction in the presence of CsF to afford the cycloadduct **10**. The TBDMS protecting group of **10** was removed with tetra-n-butylammonium fluoride (TBAF) and the compound **11** was subsequently phosphitylated with 2-cyanoethyl N,N,N’,N’-tetraisopropylphosphorodiamidite to obtain the phosphorodiamidite **12** for solid-phase synthesis (see Appendix A for detailed synthetic procedure). The vitamin E phosphoramidite was synthesized according to our previous report [26]. The benzonorbonadiene linker and vitamin E phosphoramidites were sequentially coupled to the 5′ terminal of oligonucleotides using ABI394 DNA/RNA synthesizer and vitamin E-benzonorbonadiene caged oligonucleotides was then obtained. These caged oligonucleotides were further purified by reverse-phase HPLC and characterized by ESI-MS. vitE-BL-siRNAs were prepared through the hybridization of vitamin E-benzonorbonadiene-caged antisense RNA strands with their corresponding native RNA sense strands.

To confirm the kinetics of biorthogonal reaction between vitamin E-benzonorbonadiene-caged oligonucleotides and tetrazine, we monitored the biorthogonal bond cleavage process using denaturing PAGE gel analysis. Excessive tetrazine (Tez) with different concentrations were added to the vitamin E-benzonorbonadiene-modified oligonucleotides solution and the samples were allocated at different time points. The final all solutions were loaded in PAGE gel for further electrophoresis analysis. As shown in Figure 1 and Appendix A, upon the addition of tetrazine, the cleavage of benzonorbonadiene linker led to the removal of vitamin E byproduct, and the mobility of the corresponding uncaged oligonucleotide was shifted to the same level of the native oligonucleotides. Additionally, the rate constant of the bond-cleavage reaction between caged oligonucleotide and tetrazine was calculated to be 0.073 M^−1^s^−1^, which is similar to that reported by Minghao Xu et al. [27] and was applicable for the biological applications.

### 2.2. Activation of Vitamin E-Benzonorbonadiene-Caged siRNA for GFP Gene Silencing with Tetrazine

Bioorthogonal activation of GFP gene silencing was first performed using vitamin E-benzonorbonadiene-caged siRNA in 293A cells. Native siRNA or caged siRNA were co-transfected with GFP plasmid and RFP plasmid, respectively. The GFP and RFP expression were determined using fluorescence microscopy and further quantified by flow cytometry as shown in Figure 2. The relative GFP mean fluorescence intensity was further normalized to RFP mean fluorescence intensity. The results show that the gene silencing activity of vitamin E-benzonorbonadiene-caged siRNA was nearly completely masked at 2.5 nM caged siRNA. The addition of tetrazine at a concentration that is not toxic to cells (Appendix A) significantly decreased the GFP fluorescence intensity with the corresponding RFP intensity as the internal control. However, as the concentration of caged siRNA increased, the leakage of RNAi gene silencing activity also slightly increased, while direct vitamin-E modification at the 5′ terminal of the siRNA antisense strand maintained full inhibition of siRNA gene silencing activity, similar to our previous observation [26]. These results confirm that the leakage of siRNA activity is probably due to the longer distance of the benzonorbanadiene linker between the vitamin E and siRNA. This phenomenon was similar to our previous observation that longer linker between cRGD and siRNA caused more leaking activity of siRNA gene silencing activity [28].

### 2.3. Activation of VBsiRNA for Eg5 Gene Silencing with Tetrazine

To show the generality of the biorthogonal caging strategy using inverse electron demand Diels–Alder reaction, we also performed the bioorthogonal gene silencing of the endogenous gene encoding the kinesin-5 gene (Eg5) using vitamin E-benzonorbonadiene-caged siRNA. Eg5 is essential for the assembly of the mitotic bipolar spindle and the normal mitosis processes. Inhibiting Eg5 gene expression results in the formation of monoastral spindles and cell arrest in promataphase with the increase in G2/M cells in the cell cycle [29,30]. As expected, cells transfected with native siRNA in the positive control (PC) group showed the formation of monostral spindle and the increase in G2/M cells, while either VsiEg5 (vitamin E modified siRNA without benzonorbonadiene linker) or VBsiEg5 transfection of cells had little effect on cell morphology, and the normal mitosis spindles and normal percentage of G2/M cells were observed (Figure 3). These data confirmed that vitamin E modification at the 5′ terminal of the siRNA antisense strand could effectively block the corresponding siRNA gene silencing activity. With the tetrazine addition itself in the negative control group (NC) or VsiEg5 transfection group, we did not observe these phenotypic changes, which indicated that tetrazine itself did not induce any specific cellular phenotypes and non-sensitive vitamin modified VsiEg5 also had no effect on cellular phenotypes and the cell cycle even with the addition of tetrazine. However, for cells transfected with VBsiEg5, the addition of tetrazine clearly induced the monostral spindles and increased the percentage of G2/M cells similar to that of the positive control group. These observations clearly indicated that tetrazine could react with benzonorbonadiene linker and induce the biorthogonal bond cleavage to release the active siRNA. Further qPCR quantification of Eg5 mRNA confirmed that non-sensitive VsiEg5 did not lead to the downregulation of Eg5 mRNA level with or without tetrazine addition, while tetrazine addition successfully induced the downregulation of Eg5 mRNA level in cells transfected with VBsiEg5.

## 3. Materials and Methods

### 3.1. Bioorthogonal Reaction Kinetics between Tetrazine and Vitamin E-Benzonorbonadiene-Caged Oligonucleotide

A total of 30 μL vitamin E-benzonorbonadiene-caged oligonucleotide solutions (1 μM in 1 × PBS buffer) was incubated at 37 °C. Various final concentrations (0.5 mM, 1 mM, 2 mM) of tetrazine were added to the oligonucleotide solution at different timepoints (0.5, 1, 2, 4, 6 h). The reaction solutions were finally run in 20% denaturing PAGE gel. The PAGE gels were stained with SYBR Gold nucleic acid gel stain and imaged with ChemiDoc XRS (BIO-RAD, Hercules, CA, USA). Additionally, the bands were quantified with Image J software (1.8.0, NIH, Bethesda, MD, USA)

### 3.2. GFP Gene Silencing Experiment

293A cells were obtained from National Infrastructure of Cell Line Resource (Beijing, China) and cultured in DMEM supplemented with 10% FBS, 100 μg/mL penicillin and 100 μg/mL streptomycin (37 °C, 5% CO_2_). Cells were seeded in 48-well plates with the density of 5 × 10^4^ cells/well and cultured for 24 h. To each well, 50 ng GFP plasmids (pEGFP-N1) and 100 ng RFP plasmids (pDsRed2-N1), together with siRNAs (native siRNAs or modified siRNAs), were co-transfected into cells using 0.5 μL lipofectamine2000 in optiMEM according to the standard protocol. Tetrazine (200 μM) was then added to the corresponding wells and the cells were incubated for another 6 h. The medium was replaced with fresh DMEM and the cells were further incubated for another 24 h. The cells with GFP and RFP expression were imaged using automatic inverted fluorescence microscope (Olympus, IX83, Tokyo, Japan). The excitation and emission wavelengths for GFP are 488 nm and 509 nm. The excitation and emission wavelengths for RFP are 560 and 585 nm. The cells were further digested with trypsin and collected for quantitative analysis using flow cytometer (CytoFlex, Beckman, Brea, CA, USA). The average GFP fluorescence density was normalized by average RFP fluorescence density for each experimental group.

### 3.3. Cell Cycle Analysis in Eg5 Gene Silencing Experiment

HepG2 cells were obtained from American Type Culture Collection and cultured in DMEM supplemented with 10% FBS, 100 μg/mL penicillin and 100 μg/mL streptomycin (37 °C, 5% CO_2_). Cells were seeded in 24-well plates with the density of 1 × 10^5^ cells/well and cultured for 24 h. To each well, native siRNAs or modified siRNAs targeting Eg5 were transfected into cells using 1 μL lipofectamine2000 in optiMEM according to standard protocol. Tetrazine (200 μM) was then added and the cells were incubated for another 6 h. The medium was replaced with fresh DMEM and the cells were further incubated for another 36 h. The cells were digested by trypsin, washed with 1 × PBS buffer and collected. The cells were fixed with 75% ethanol at 4 °C overnight. Then, the cells were centrifugated (1000× *g*, 5 min), washed with 1 × PBS buffer and finally resuspended in 400 μL 1 × PBS buffer, followed by further incubation with propidium iodide (PI) and RNase A solution at 37 °C for 30 min in the dark. The PI fluorescence of cells was analyzed with flow cytometer (CytoFlex, Beckman, Brea, CA, USA). The data were fitted and the percentages of cells in G2/M phase were obtained with ModFitLT software (Verity Software House, Topsham, ME, USA).

### 3.4. qRT-PCR of Eg5 mRNA in Eg5 Gene Silencing Experiment

HepG2 cells were seeded in 12-well plates with the density of 2 × 10^5^ cells/well and further cultured for 24 h. The cells were treated according to the above cell cycle for all the experimental group after transfection of the corresponding siRNAs. The total RNA was extracted using BIOZOL Total RNA Extraction reagent and was quantified with NanoDrop 2000. The RNAs were reversely transcribed to cDNAs using HiScript II Q RT SuperMix for qPCR. The quantitative RT-PCRs of Eg5 gene expression level (Forward primer: CAGCTGAAAAGGAAACAGCC, Reverse primer: ATGAACAATCCACACCAGCA) were performed with GAPDH mRNA as the internal control (Forward primer: TGCACCACCAACTGCTTAGC, Reverse primer: GGCATGGACTGTGGTCATGAG).

### 3.5. Cell Phenotypes in Eg5 Gene Silencing Experiment

HepG 2 cells were seeded in 4-chamber 35mm glass bottom dishes with density of 1 × 10^5^ cells/chamber and were cultured for 24 h. The cells were treated according to above cell cycle for all the experimental group after transfection of the corresponding siRNAs. Then, the cells were fixed with 200 μL Immunol Staining Fix Solution. After 10 min of incubation, the cells were washed with 400 μL Immunol Staining Wash Solution 3 times and then stained with Tubulin-Tracker Red. After 45 min incubation, the cells were washed with PBS buffer and stained with Hoechst 33,342 for 5 min. The cells were further washed with PBS buffer 3 times and were then imaged using laser scanning confocal microscope (Zeiss LSM880) at excitation wavelength of 561 nm for Tubulin-Tracker Red and 405 nm for Hoechst 33,342.

## 4. Conclusions

In summary, we rationally designed and synthesized a new kind of bioorthogonal caged siRNA with vitamin E modification at the 5′ terminal of the antisense strand through a benzonorbonadiene linker. The benzonorbonadiene linker could be efficiently cleaved though biorthogonal inverse electron demand Diels–Alder (IEDDA) bond cleavage reaction upon the addition of tetrazine. We evaluated the biorthogonal activation of these vitamin E- benzonorbonadiene-caged siRNAs for conditional gene silencing of both exogenous EGFP and endogenous Eg5 genes with tetrazine. The results indicate that vitamin-E-benzonorbonadiene modification could temporally mask the siRNA gene silencing activity, which could be effectively restored upon the addition of tetrazine. Our data might provide a new strategy of “bioorthogonal caging siRNA” using biocompatible chemical reactions for studying gene functions and developing nucleic acid prodrugs

## Data Availability

The data presented in this work are available in the article and Appendix A.

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
