# Peer review of "Tetrazine-Induced Bioorthogonal Activation of Vitamin E-Modified siRNA for Gene Silencing"

_molecules, 2022, doi:10.3390/molecules27144377_

Round 1
Reviewer 1 Report
Small interfering RNA (siRNA) has been emerging as a powerful gene silencing tool and innovative nucleic acid medicines for its sequence-specific degradation of mRNA. Conditional activation of siRNAs offers another layer of control for this technology. In this work, the authors chose vitamin E-benzonobonadiene-caged-siRNA as the proof-of-concept study to demonstrate the potential application of the biorthogonal uncaging strategy for controlling the siRNA activity. These chemical stimuli responsive vitamin E-caged siRNAs are designed to target both green fluorescent protein (GFP) gene and mitotic kinesin-5 (Eg5) gene. They have been synthesized through a benzonobonadiene linker between Vitamin E and the 5’-terminal of siRNA antisense strand. Upon the activation of tetrazine, the biorthogonal reaction induced the breakage of benzonorbonadiene linker and released the active siRNAs for the gene silencing of both endogenous and exogenous genes in cells. Although there have been quite a few caging methods available to control siRNA activity, this new bioorthogonal IEDDA reaction-based approach is very attractive for in vivo application due to its fast reaction rate and low cellular toxicity of tetrazine. The experimental data are solid and the manuscript is well written, I strongly recommend accept this paper in its current form.
Author Response
We thank this reviewer very much for your comments. We have proofread our manuscript as possible as we can.
Reviewer 2 Report
The manuscript by X. Zhang and co-workers describes the chemical synthesis of benzonorbonadiene-caged siRNAs that are activated by the addition of triazine. The authors demonstrated this chemical activation in two genes: a reporter gene that is introduced into cells by plasmid transfection (GFP), and an endogenous gene (Eg5). Altogether, the strategy is a nice addition to uncaging strategies used thus far for activating gene silencing of oligos (siRNA and ASO).
I would consider publication in Molecules, however, at the current state, the manuscript is lacking many experimental details that without their addition it would be impossible to reproduce the data reported in this manuscript.
Thus, the authors should provide the following details in the ESI:
11. Please provide MS for compounds 2-4
22. Please provide yields for compounds 7 and 8.
33. Page 12 – please provide amounts for compound 8. And amounts for 0compounds 4, 9, and CsF.
44. Page 13 – please provide volume of THF used and amount of TBAF. Also provide amount of compound 11, tetrazole, and phophoroamidite. Add volume of DCM used.
55. Include the HPLC column type (brand) used.
66. Add primer sequences used for qRT-PCR for both genes (page 7 of MNS).
In addition, the English in the manuscript should be improved. I am giving some examples for page 1:
aa. Glutataione- change to glutathione
bb. Cellular study – change to cellular systems
cc. Readily to be synthesized – remove "to be"
dd. et al – change to "and others"
ee. change to -their fast reactions rates
f.f. induced – change to provided
Other points in MNS:
11. Figure 2 – Add Vitamin E as one of the released products
22. Figure 2 – Concentrations of tetrazine shown in plot C do not correlate with those shown in the gel (B)
33. There is no reference to Figure 3 in the MNS text
44. Statistics are not given for figures 3 and 4 – how many repeats? Standard deviation?
55. The authors should add in the MNS text that tetrazine is not toxic at the concentrations used for cell experiments. This is shown only later in the ESI.
Author Response
We thank this reviewer very much for your usefully comments. We have made the changes according to the reviewer's suggestions and listed the responses as following:
the authors should provide the following details in the ESI:
- Please provide MS for compounds 2-4
Response: Thank you for your comments. Actually all these compounds are previously reported and we synthesized according to the literature (Minghao Xu et al., Chem. Commun., 2017,53, 6271-6274, no mass spectra were provided also). Due to the very simple compounds, we could easily identify these compounds based on HNMR and CNMR. And the student did not characterize them using MS spectra, but we did characterize the final products using these intermediates.
- Please provide yields for compounds 7 and 8.
Response: Thank you very much for your comments. We are sorry for this missing information. We have included these information in ESI.
- Page 12 – please provide amounts for compound 8. And amounts for 0compounds 4, 9, and CsF.
Response: Thank you very much for your comments. We are sorry for this missing information. We have included these information in ESI.
- Page 13 – please provide volume of THF used and amount of TBAF. Also provide amount of compound 11, tetrazole, and phophoroamidite. Add volume of DCM used.
Response: Thank you very much for your comments. We are sorry for this missing information. We have included these information in ESI.
- Include the HPLC column type (brand) used.
Response: Thank you very much for your comments. We have added the information in the manuscript.
- Add primer sequences used for qRT-PCR for both genes (page 7 of MNS).
Response: Thank you very much for your comments. We have added the information in the manuscript.
In addition, the English in the manuscript should be improved. I am giving some examples for page 1:
- Glutataione- change to glutathione
- Cellular study – change to cellular systems
- Readily to be synthesized – remove "to be"
- et al – change to "and others"
- change to -their fast reactions rates
- induced – change to provided
Response: Thank you very much for your comments. We tried our best to proofread the manuscript and correct these mistakes including the references.
Other points in MNS:
- Figure 2 – Add Vitamin E as one of the released products
Response: Thank you very much for your comments. We made the changes and added the released vitamin E by-product in Figure 2.
- Figure 2 – Concentrations of tetrazine shown in plot C do not correlate with those shown in the gel (B)
Response: Thank you very much for your suggestion. We corrected the error of tetrazine concentration in plot C.
3.There is no reference to Figure 3 in the MNS text
Response: Thank you very much for your suggestion. We added the reference to Figure 3 in the MNS text.
- Statistics are not given for figures 3 and 4 – how many repeats? Standard deviation?
Response: Thank you very much for your suggestion. All the experiments were done at lease 3 time. And we added statistics description in the caption of Figure 3 and Figure 4.
- The authors should add in the MNS text that tetrazine is not toxic at the concentrations used for cell experiments. This is shown only later in the ESI.
Response: Thank you very much for your suggestion. We included the toxic property of tetrazine in ESI as Figure S2 and inserted the reference in section 2.2 of the MNS text.